# Ivory Coast without ivory: Massive extinction of African forest elephants in Côte d'Ivoire

Jean-Louis Kouakou[1], Sery Gonedelé Bi[1,2]*, Eloi Anderson Bitty[1,2],
Célestin Kouakou[2,3‡], Alphonse Kouassi Yao[1], Kouadio Bénoîtype Kassé[1‡],
Soulemane Ouattara[1]

**1** Laboratoire des Milieux Naturels et Conservation de la Biodiversité, Université Félix Houphouët Boigny d'Abidjan-Cocody, Abidjan, Côte d'Ivoire, **2** Centre Suisse de Recherches Scientifiques en Côte d'Ivoire, Abidjan, Côte d'Ivoire, **3** Unité de Formation et de Recherche d'Environnement, Université Jean Lorougnon Guédé, Daloa, Côte d'Ivoire

☯ These authors contributed equally to this work.
‡ CK and KBK also contributed equally to this work.
* sery.gonedele@univ-fhb.edu.ci

## Abstract

In pre-colonial and colonial times Côte d'Ivoire probably hosted one of the largest elephant populations in West Africa, resulting in the country's name Côte d'Ivoire (in English Ivory Coast) by French settlers. Numbers declined and by the early 90s it was estimated that the total number of both savannah and forest elephants had reached 63 to 360 elephants in the entire country. Here we present updated information on the distribution and conservation status of forest elephant in Côte d'Ivoire based on multiple sources—dung counts on line transects, records of human–elephant conflict, media reports, sign and interview surveys—obtained during the period 2011–2017. We used Pearson correlation to determine the correlation between the presence of forest elephant and site variables (size of the forest, percentage of area converted into plantation, size of the forest left, size of human population inside the PA, poaching index, distance to the nearest road, population density in the Department, level of protection of the PA). To examine the effect of ecological traits on elephant extirpation, we used Principal Components Analysis (PCA) to check for multicollinearity among variables. Based on dung count elephant presence was confirmed in only 4 of the 25 protected areas surveyed. PAs with higher level of protection have higher probability to be home of elephant population. The viability of these populations is uncertain, since they have a small size and are isolated. Aggressive conservation actions including law enforcement for the protection of their remaining habitat and ranger patrolling are needed to protect the remaining forest elephant populations.

## Introduction

Africa is home to at least 400,000 elephants, an estimated 5,458 of which are found in West Africa [1]. Genetic evidence suggests two distinct species among African elephants: the forest elephant (*Loxodonta cyclotis*) and the savanna elephant (*L. africana*) [2, 3].

**Data Availability Statement:** All relevant data are within the manuscript and its Supporting Information files.

**Funding:** The author(s) received no specific funding for this work.

**Competing interests:** The authors have declared that no competing interests exist.

Among the large mammals of Africa, the elephant is probably one of the most affected by human activities. The greatest threat to them in West Africa comes from habitat loss due to human encroachment [4]. Elephant habitat in West Africa has been fragmented, most of which are subjected to significant pressures from human population growth, the spread of agriculture, and livestock production surrounding the parks [4].

In pre-colonial times (before 1893) Côte d'Ivoire probably hosted one of the largest elephant populations in West Africa. Despite the early harmful effects of human activities, elephant numbers remained high until colonial time in West Africa, resulting in the country's English name—Ivory Coast—given by the colonial powers to this area [5]. However, during the three last three decades, elephant populations have been sharply reduced, mainly because of forest agricultural clearing [6]. Roth [7] estimated a total population of 1790 individuals in savannah and 3050 forest elephants scattered throughout Côte d'Ivoire.

Numbers declined further by the late 1980s when [8] presented data on the remaining 20 isolated populations of forest elephants. There was a further 50% decline in numbers. The early 90s saw even further declines when the total number of both savannah and forest elephants dropped to 63 to 360 individuals respectively, in 24 populations across the entire country [1]. These populations are confined to protected areas and continue to decline. Indeed, at the outset of the 20th century, there were sixteen million hectares of high canopy forest existed in Côte d'Ivoire; today, that number is four millions ha and declining due to an annual deforestation rate of approximately 1% [9, 10]. The remaining forest of Côte d'Ivoire is highly fragmented and largely consists of nominally protected national parks and forest reserves. Wildlife in these protected areas is threatened by hunting, the encroachment of cocoa plantations on reserve borders, and expansion of illegal cocoa farming within the parks and reserves themselves [11, 12].

Elephant numbers in Côte d'Ivoire has decreased at an alarming rate since the first population surveys were conducted, and warnings that this species might be lost were put forward in 1984 [13]. The situation is probably even worse, with only seven elephant populations confirmed in the last decade, comprising approximately 270 elephants [13]. The outbreak of civil war in Côte d'Ivoire in September 2002 negatively affected the management of certain protected areas and conservation in general [14], and although not quantified it can be assumed that the effect on the small population of elephants was negative if not catastrophic [13].

Within the scope of the national strategy for elephant conservation in Côte d'Ivoire, the International Union for Conservation of Nature [15] recommended the need for a survey of the main habitats for elephants and the identification of viable populations. The most recent data on the population of Côte d'Ivoire elephants are at least one decade ago and most of these data did not follow a standardized protocol. The aim of this study is to obtain updated information on the distribution and conservation status of the forest elephants of Côte d'Ivoire.

## Methods

### Study sites

We used historical distribution data and information from elephant surveys to identify twenty-six protected areas (PA) in central and southern Côte d'Ivoire known to contain elephants (Fig 1). The surveyed localities belong as well as to the Guinean forest zone below the V-Baoulé in the southern part of Côte d'Ivoire. Protected areas are categorized into National Parks, which normally are under strong protection restriction, and Forest Reserves that are partially protected. We selected the PAs to be surveyed based on the presence of forest habitat. The exception were Marahoué National Park and Koba Forest Reserve in central Côte d'Ivoire. These both consist of both dense forest and savanna woodland.

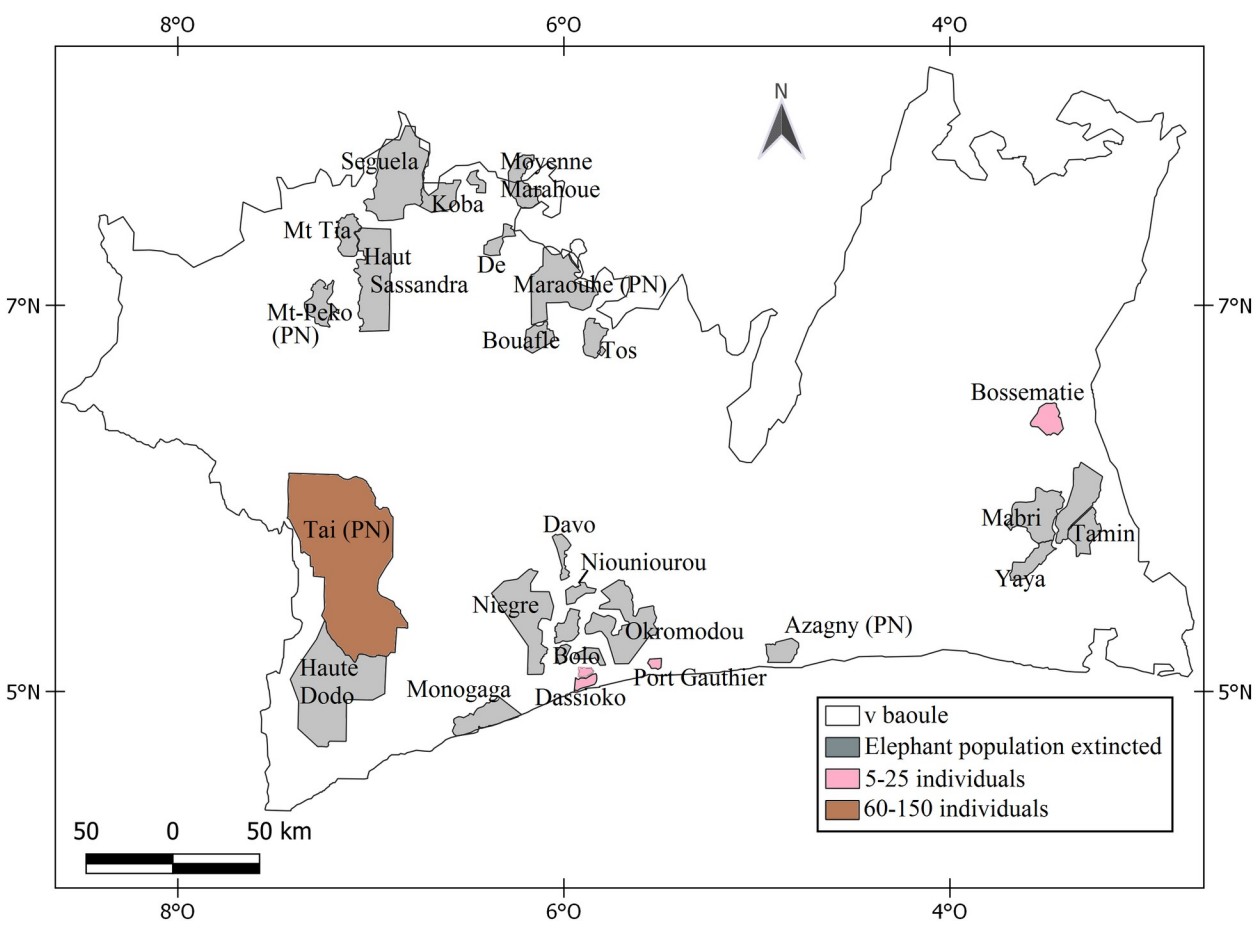

**Fig 1. Location of the surveyed localities in Côte d'Ivoire.**

Prior to the surveys, we received research permits from SODEFOR (Society of Forest Development) and OIPR (Ivorian Office of Parks and Reserves), respectively in charge of the management of Côte d'Ivoire's forest reserves and parks. SODEFOR and OIPR reviewed and approved the study.

## Assessment of forest elephant distribution

From 2011–2017, we collected data on elephant occurrence from multiple sources including dung counts along line transects, records of human-elephant conflict, media reports, sign, and interview surveys. We also collected data on elephant distribution through structured interviews.

Interviews were conducted with key informants from the surrounding local communities of PAs as well as field staff of the Forest and Wildlife Department. Before the interview, we explained the research purpose to the informants. We received a verbal consent of their acceptance to participate in the study. We gathered data on locations of elephant in news reports from newspapers and Internet searches for verifiable news reports of elephant presence in localities close to PAs where elephants' presence was confirmed. We also talked to researchers who conducted scientific or conservation work in these areas and the bordering areas.

We focused mainly on dung counts on line transects carried out across eight surveyed sites (Dassioko Sud FR, Port Gauthier FR, Monogaga FR, Niégré FR, Okroumodou FR, Azagny NP,

Bolo Ouest, Marahoué NP) during the wet period (from April to July) and the dry one (from December to March) of 2011 to 2017.

The PAs surveyed extend in the southern zone of Côte d'Ivoire, the historical range of forest elephants, thus we assume that the population surveyed are forest elephants.

We made efforts to place line-transects within each forest where elephants were known to occur. The field protocol involved surveyors in a team of two persons walking once along each transect of 2–9 km length. When elephant dung piles were encountered, their number, and the perpendicular distance from transect to the center of the dung piles were recorded in addition to data on vegetation type [16, 17]. These data were used to infer elephant distribution but also to compute dung densities for each transect (in dung piles/ha), which we treated as an index of elephant density.

During surveys, we also recorded all signs of poaching (e.g., number of gunshots heard, snares, discarded cartridges, etc.) and used these to calculate a poaching index, which is defined as the number of poaching signs per kilometer walked in each PA.

### Data analysis

Data collected on the presence of forest elephant from various sources were overlaid on the map of forested areas, including PAs and non-PAs. Dung density was computed for each transect as the ratio of dung piles encountered on transect to the area of a rectangular quadrat described by the product of transect length and twice the average perpendicular distance (as strip width) [18, 19]. For multiple transects, we obtained a point value of dung density by summing all dung piles encountered and dividing by the total area covered by all transects.

In the first step, the correlation between the presence of elephant and site variables (size of the forest, percentage of area converted into plantation, size of the forest left, size of human population inside the PA, poaching index, distance to the nearest road, population density in the Department, level of protection of the PA) was evaluated with Pearson correlation. These variables were log transformed as needed to improve normality.

To examine the effect of ecological traits on elephant extirpation, we used Principal Components Analysis (PCA) to check for multicollinearity among variables [20]. Second, to generate a pool of candidate multi-regression models, we sequentially removed, from the full model, the variables with the smallest contribution to model adequacy, based in the decrease of log-likelihood value generated by logistic regression of models when each variable was removed. To evaluate which combination of variables best fitted our data, we compared models with all possible combinations of main effects based on the second-order model selection criteria (AICc) [21, 22]. The lowest AICc value indicates the model that achieves the best tradeoff between variance and bias of the resulting parameter estimates. In addition to calculating AICc values, Akaike weights ($wi$) were calculated [21]. Weights sum to 1 and provide a measure of the weight of evidence in favour of one model over the others [23]. We also calculated $\Delta i$, as the difference between the AICc for the $i$th model in the set and the minimum AICc. Delta values can be used to gauge the relative plausibility of each model.

Statistical analysis was conducted in SPSS statistical software version 17.0 [24].

## Results

### Elephants censured

Based on dung count elephant presence was confirmed in 4 of the 25 protected areas surveyed. Moreover, outside of the surveyed protected areas, conflict data and media reports confirmed the presence of elephants in 4 localities (Daloa, Dassioko, Belleville and Sikensi). Hence four (57.1%) of the seven sites where elephants were observed are protected areas whereas the 3

others (42.9%) are out of protected areas. Of the four PAs, density estimates were based on line-transect survey in three PAs (Dassioko Sud, Port Gauthier FR and Bossematié FR), whereas for one of these PAs (Taï NP), density estimates were based on previous reports from researchers. Our results show a widespread and catastrophic decline in the occurrence and numbers of forest elephants. Of the 25 PAs surveyed, 21 (84%) have lost their entire population of forest elephant. Of the four (16%) PAs where elephant populations have survived, dung count indicates a low density (0.1197 to 0.168 individual/km$^2$) of elephant within these forests. While being evidently absent from some nominally large PAs, elephants still occurred in small number (4 to 8 individuals) outside PAs in areas with little or no forest cover at the vicinity of PAs that have lost their elephant population. This is the case of the individuals found in Daloa, Belleville and Sikensi, respectively at the vicinity of Haut Sassandra FR and Marahoué NP, Mont Péko NP. The five individuals found near the village of Dassioko were formerly translocated population that comes from Petit Bondoukou. The two individuals localized in Sikensi were recently translocated in April 2015 from Daloa to Azagny NP during a risky and expensive process.

## Threat on forest elephants

More than half of the protected areas surveyed (Bolo-Ouest FR, Niégré FR, Monogaga FR, Rapide Grah FR, Haute Bolo FR, Marahoué National Park, Okroumodou FR, Koba FR, Davo FC and Duékoué FR) have been completely converted to farms and human settlement. A significant and positive correlation was found between the presence of elephants in the PAs and the level of protection of the PA (r = 0.804, p < 0.0001, n = 25). A significant and negative correlation was found between the presence of elephants and human population inside the PA (r = - 0,174, p = 0.019, n = 25. No significant correlation was found between the presence of elephants and the other variables (Fig 2): eg. presence of elephant's *vs* proportion of area converted into plantation (r = - 0.271, p = 0.190, n = 25), presence of elephant's *vs* distance to the nearest road (r = - 0.176, p = 0.399), presence of elephant's *vs* size of the PA (r = 0.217, p = 0.298), presence of elephant's *vs* poaching index (r = 0.223, p = 0.283).

## Multivariate analysis

A PCA was used to determine how the variables (size of the forest, percentage of area converted into plantation, size of the forest left, size of human population inside the PA, poaching index, distance to the nearest road, population density in the Department, level of protection of the PA) affect the presence/absence of elephant population in the PAs surveyed.

The first two axes of the PCA matrix (Fig 3) account for 70% of the variation in the data, with eigen values of λ1 = 2.37 and λ2 = 1.22, respectively. PCA Axis 1, which accounted for 46.22% of the variation, was highly correlated with anthropogenic variables (human population size, habitat degradation, proportion of forest converted to cocoa plantation).

The principal component ordination of the presence of elephant population over the nine variables clearly distinguish PAs where forest fragment has left with no population settlement inside (Bossématié FR, Azagny NP, Dassioko Sud FR, Port Gauthier FR) and PAs where the forest have been cleared for agriculture purpose and where human population have been settled (Bolo Ouest FR, Marahoué NP, Monogaga FR and Mont Péko NP) (Fig 3). This suggests that the extent of forest elephant is forest and human population settlement inside the PAs dependent. As traditional test statistics may be confounded by collinearity, we selected the most parsimonious model from a set of 26 possible combinations of independent variables, based on AICc values (Table 1). Of the three candidates' models generated based on the contribution of each variable to model adequacy, the most parsimonious model selected by AICc

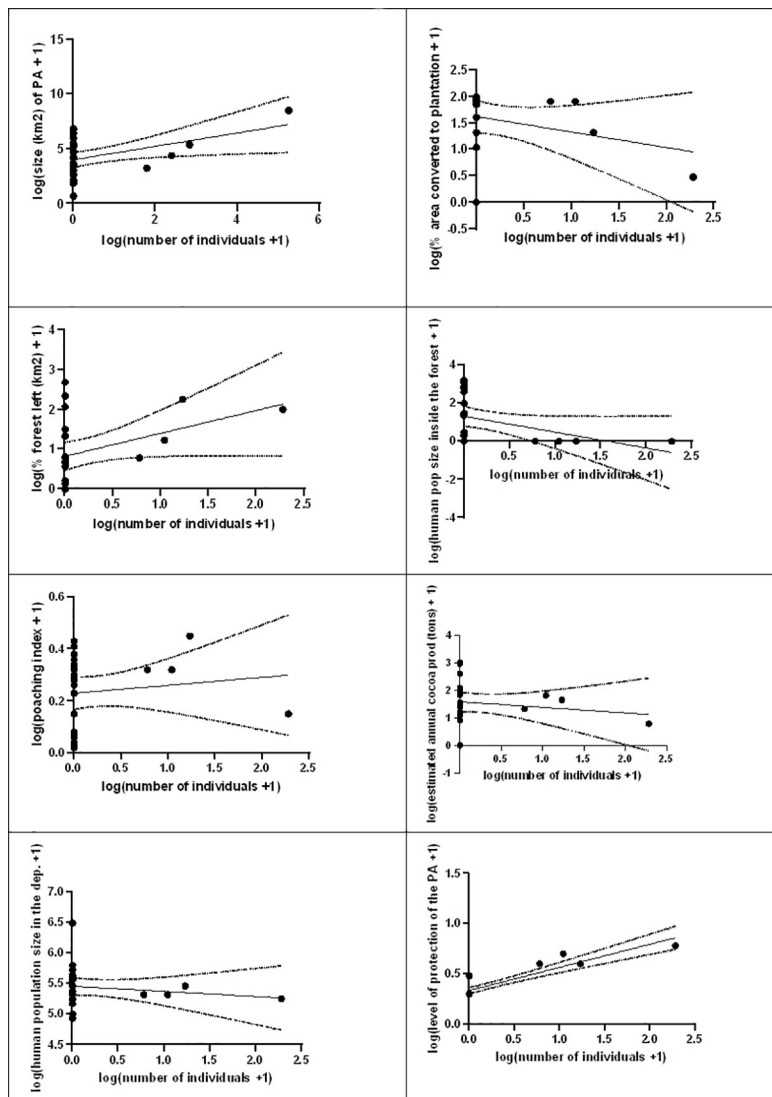

**Fig 2. Correlates between the presence of elephants in the PAs and environmental variable (size of the forest, percentage of area converted into plantation, size of the forest left, size of human population inside the PA, poaching index, distance to the nearest road, population density in the Department, level of protection of the PA).**

was one that included the level of protection of the PAs (Table 2). Thus, in the final model, PAs with higher level of protection have higher probability to inhabit elephant population. This result confirms that, the level of protection of PAs was the pillar of the survival of the population of African forest Elephants in Côte d'Ivoire.

## Discussion

The dramatic decline of forest elephant populations in Côte d'Ivoire is primarily attributed to the lack of protection of the PAs that remains their only refuge. However, increasing human density around the PAs, encroachment of the PAs, and increasing human activities inside the PAs are decisive additional factors enhanced by the lack of protection.

Over the two latest decades, the population of the forest elephants in Côte d'Ivoire has been reduced by 90%. In the same time, the number of PA harboring forest fragment fell by 80%.

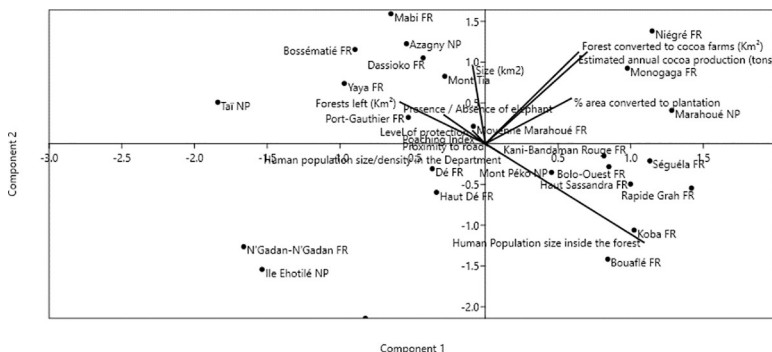

**Fig 3. Biplot of principal coordinate analysis of the presence of forest elephants in 25 PAs correlated with environmental variable (size of the forest, percentage of area converted into plantation, size of the forest left, size of human population inside the PA, poaching index, distance to the nearest road, population density in the Department, level of protection of the PA).** Coordinates 1 and 2 explain 70% of variation in the data.

The large majority of the PAs has lost its entire elephant populations as a consequence of the lack of conservation measures, conversion of protected areas into plantations, human settlement and poaching inside the PAs.

## Population decline

In Côte d'Ivoire, the forest elephant population has risen from 1,611 in 1994 [29] to 225 today, representing a population decline of 86% during the latest decades.

The complete absence of records of elephant population in the PAs surveyed indicates that their declines have occurred throughout their geographic range. The absence of records does not seem to be an artifact of low search effort. A considerable portion of the range of forest elephant surveyed have been accessed and repeatedly visited by our team. The available evidence suggests that forest elephants have been extirpated in PAs where they were not observed and that their population has sharply decline in PAs where they were observed.

Also, there are reports of migrations of elephants that resulted in the occupation of the PAs by cocoa plantations and human settlement inside PAs such as Marahoué NP, Mont Péko NP and Niégré FR. Civil unrest, human encroachment, and habit fragmentation leaves some elephants confined to small patches of forest without sufficient food [30]. As a result, elephant populations have been recorded in human disturbed areas, such as Belleville (around Mont Péko NP), Daloa (aroud Marahoué NP and Haut Sassandra FR). At the request of Côte d'Ivoire's government, during a risky process 6 individuals were translocated from Daloa to Azagny NP in 2014. Two of them died during the translocation. Of the four individuals successfully translocated in Azagny NP, two individuals fled from Azagny NP and were retrieved in Tiassalé.

This study adds to a growing list of studies that document declines of large mammals within protected areas across the continent [31, 32]. Given the current level of protection of the protected areas of Côte d'Ivoire's and available range of forest elephant's occurrence, their survival in the near future is uncertain.

**Table 1. The candidate model set and AICc rankings for the presence/absence of forest elephant population in the PAs in Côte d'Ivoire.**

| Model | Variables | AICc | Δi | r² | p |
|---|---|---|---|---|---|
| 1 | Level of protection | 48 | 2 | .675 | < .0001 |
| 2 | Level of protection, Human Population size inside the forest | 38 | 10 | .770 | < .0001 |
| 3 | Level of protection, Human Population size inside the forest, Population density in the Department (Hbts/km2) | 46 | 12 | .824 | < .0001 |

**Table 2. Trend of African forest elephant (*Loxodonta africana cyclotis*) occurrence in Côte d'Ivoire.**

| Forests | Population trend | | Methods | Sources |
|---|---|---|---|---|
| | 1991–2003 | 2011–2017 | | |
| Bolo-Ouest FR | 10 (1991) | 0 | Ground census (transects) | [8]; present study |
| | | | Line transects, dung count | |
| Niégré FR | 50 (1991) | 0 | Ground census (transects) | [8; 25]; present study |
| | | | Line transects, dung count | |
| Monogaga FR | | 0 | Line transects, dung count | present study |
| Marahoué NP | 50 (1991) | 0 | Informed guess | [1] ; present study |
| | | | Line transects, dung count | |
| Rapide Grah FR | 70 (1991) | 0 | Ground census (transects) | [8] ; present study |
| | | | Line transects, dung count | |
| Haut Sassandra FR | 30 (2002) | 0 | Informed guess | [26] ; present study |
| | | | Line transects, dung count | |
| Mont Péko NP | 20 (1991) | 0 | Ground census (transects) | [8] ; present study |
| | | | Line transects, dung count | |
| Bouaflé FR | - | 0 | Line transects, dung count | present study |
| Koba FR | - | 0 | Line transects, dung count | present study |
| Dé FR | - | 0 | Line transects, dung count | present study |
| Haut Dé FR | - | 0 | | present study |
| Moyenne Marahoué FR | - | 0 | Line transects, dung count | present study |
| Azagny NP | 65 (2003) | 0 | Line transects, dung count | present study |
| Bossématié FR | 30 (1994) | 16 | Informed guess | present study |
| Mabi FR | 5 (1994) | 0 | Informed guess | [27] ; present study |
| Yaya FR (Yaya-Mabi-Songon-Tamin complex) | 5(1994) | 0 | Informed guess | [27], present study |
| Taï NP | 160 (2011) | 189 | Ground census (transects) | [28] |
| Okromodou FC | 50 (1991) | 0 | Ground census (transects) | [8] ; present study |
| | | | Line transects, dung count | |
| Davo FC | 20 (1991) | 0 | Ground census (transects) | [8] ; present study |
| | | | Line transects, dung count | |
| Duékoué FR | 15 (1976) | 0 | Ground census (transects) | [8] ; present study |
| Dassioko Sud FR | | 10 | Line transects, dung count | present study |
| Port-Gauthier FR | | 5 | Line transects, dung count | present study |

Our analysis identified several factors likely to contribute to this decline and demonstrated the importance of law enforcement for persistence of elephants. Similar factors were also found to be important in recent analyses of a very different dataset- carcass data from the MIKE sites [33, 34]. Higher levels of elephant poaching, as expressed by the proportion of illegally killed elephants (PIKE) were associated with sites where law enforcement capacity was lower, and in countries with poor governance.

## Weakness of conservation efforts

The lack of effective governance facilitates the conversion of Côte d'Ivoire's PAs into plantation and human population settlements. The four PAs where elephant populations have survived have received a kind of protection compared to the other PAs. For Taï NP, the forest received international supports even during the civil war period [35]. Dassioko and Port Gauthier received periodic support for surveillance activities from 2011 to 2017 with the involvement of local communities into the surveillance activities [36]. During this period, which included a civil war (2002–2004), hundreds of thousands of persons moved into central and

southern Côte d'Ivoire from other portions of the country and from neighboring Mali and Burkina Faso, with many migrants taking up residence adjacent to or within forest reserves and national parks [14]. Given the government's concerns with national security, safeguarding habitat and wildlife inside parks and forest reserves was likely not a high priority, and thousands of migrants readily occupied protected areas. Most conservation staff charged with monitoring, protecting fled their parks/reserves, and south-moving migrants encountered little if any resistance. The result was the rapid establishment of permanent human settlements, an increase in cocoa farming, and an escalation of hunting within the country's protected areas [10, 37].

Most of the PAs in Côte d'Ivoire do not receive any tangible wildlife management, and the conservation status of forest elephants is considerably worse. Forest elephants will be extinct in Côte d'Ivoire unless immediate actions are implemented to safeguard the remaining population. Indeed, the neighboring country, Ghana has demonstrated that with improved strategies for law enforcement, including shop raids on ivory and monitoring the work of patrol staff in protected areas, elephant populations can be better secured [38].

However, despite massive declines in numbers, our study has shown that the commitment of highly motivated government field staff, and the continued support by international organizations to provide some protections on the ground, made a difference for their survival as revealed by the survival of forest elephants in protected areas such as Dassioko Sud and Port Gauthier. In sites where this protection could not be provided, elephant populations have been extirpated. Therefore, even limited efforts to invest in conservation during periods of political turmoil have benefits for biodiversity [39, 40]. There have been similar observations in Rwanda, where parks and reserves that received support from international NGO's were far less affected by the genocide of 1994 than sites with no support [41, 42].

## Habitat loss

The 25 forests surveyed totalize 3688.66 km$^2$, of which 71% (2611.98) have been cleared or transformed into plantation. Côte d'Ivoire has the highest deforestation rate in sub-Saharan Africa, with an estimated loss of 265,000 ha per year [11]. Little primary forest exists in south central Côte d'Ivoire, even within protected areas, and that which remains is at risk of being replaced by agricultural plots [14]. Hence reducing the habitat and food resources for the highly frugivorous African forest elephants [43–46] and other animals.

Protected areas destruction has also been a dominant factor shaping population change and distributions of the Côte d'Ivoire's forest elephants in the last two decades. Habitat loss can have a profound impact not only on species numbers, but also on community functioning [47, 48]. The simplification of habitats leads to the simplification of trophic food webs [49]. As a consequence of their habitat transformation/lost, several elephant population fled outside PAs. This increased human-elephants conflict outside PAs, hence exposing elephant to poaching.

The remaining populations of the forest elephants of Côte d'Ivoire are now isolated in PAs surrounded by agriculture plots as already reported in other countries [50, 51].

With the rapid expansion of cocoa plantations, forests were cut down and the elephant range in the forests was further reduced. Human population growth spread into elephant areas and human-elephant conflict increased; farmers retaliated against crop raiding by killing elephants. Sporadic elephant poaching have certainly feed ivory market in Côte d'Ivoire. Indeed, Côte d'Ivoire was a major market for ivory trade in West Africa [52] and some of Ghana's tusks reached Abidjan, either for carving or for re-export.

With the rapid growth of human population in Côte d'Ivoire, people require more food and space which requires more land for agriculture and habitation [53]. During the last two

decades, significant portions of Côte d'Ivoire PAs have been modified by human activities. Several studies [54, 55] have reported that this loss is particularly acute with large species.

## Conservation status

Currently the Red List classifies African forest elephants (*Loxodonta africana cyclotis*) as Vulnerable [56]. The actual list represents the status of these taxa at a global level and cannot be applied at the Côte d'Ivoire level with regard to the threats that they are facing in this country. Our present data indicated that the forest elephant in Côte d'Ivoire been locally extirpated in 80–95% of the reserves where they were reported a decade ago. Indeed, they underwent an observed, estimated, inferred or suspected population size reduction by > 80% and a decline in their extent of occurrence and their area of occupancy over the last 10 years. For the remaining forests where they actually occurred, their population size seems to have sharply declined so that their viability remains questionable. Regards to the actual trend of the forest elephant in Côte d'Ivoire, the species should be listed as Critically Endangered. Indeed, the criterion for listing a species as Critically Endangered is when that species has declined by 80% in ten years, or three generations, whichever is the longer.

The lack of updated data on the population trend of Côte d'Ivoire's forest elephant has certainly contributed to the actual threat. Indeed, the publish of relevant research into the public domain through timely publication, is baseline to inform conservation community to react, in order to judge and react to trends before any more disasters occur.

## Conclusion

The range of the forest elephants in Côte d'Ivoire has increasingly been reduced in the latest decades and is now limited to four PAs and three unprotected areas, increasingly isolated due to changes in land use. For many reasons the viability of these populations is questionable, since their size is largely reduced and their populations are isolated. Aggressive conservation actions including law enforcement for the protection of their remaining habitat and anti-poaching actions are needed to protect the remaining forest elephant populations. For the populations of Dassioko Sud and Port Gauthier FR located in the same areas, since their long-term viability in small isolated population is unsure, there is a need to establish a corridor between the two forest patches to unsure genetic changes. The success of such corridor establishment will need the involvement of local people, since the areas between these PAs is now largely converted into plantations. With the increasing human pressure around the remaining PAs, human-elephant conflicts will inevitably increase. That suggests that Ivoirian authorities have to initiate innovative strategies involving local communities, to avoid such conflict that often results in death for either human or elephants and economic lost.

## Supporting information

**S1 Appendix. Data collection sheets.**
(PDF)

**S2 Appendix. Elephant survey questionnaire.**
(PDF)

**S3 Appendix. Supporting data.**
(PDF)

**S1 File.**
(PDF)

**S2 File.**
(PDF)

**S3 File.**
(PDF)

**S4 File.**
(PDF)

**S5 File.**
(PDF)

**S6 File.**
(PDF)

## Acknowledgments

We would like to thank the Minister of Scientific Research of Côte d'Ivoire for the research permit and the Swiss Centre of Scientific Research for the logistical support that they offered. We are also grateful to the SODEFOR (Société de Développement des Forêts) and the OIPR (Office Ivoirien des Parcs et Réserves) for the permit to access the protected areas. We thank the local communities around the surveyed forests and the field guides for their assistance.

## Author Contributions

**Conceptualization:** Jean-Louis Kouakou, Sery Gonedelé Bi, Eloi Anderson Bitty.

**Data curation:** Jean-Louis Kouakou, Sery Gonedelé Bi.

**Formal analysis:** Jean-Louis Kouakou.

**Investigation:** Jean-Louis Kouakou, Sery Gonedelé Bi, Eloi Anderson Bitty, Alphonse Kouassi Yao, Kouadio Bénoîtype Kassé.

**Methodology:** Jean-Louis Kouakou, Sery Gonedelé Bi, Eloi Anderson Bitty, Célestin Kouakou, Alphonse Kouassi Yao, Soulemane Ouattara.

**Software:** Célestin Kouakou.

**Supervision:** Sery Gonedelé Bi.

**Validation:** Sery Gonedelé Bi, Soulemane Ouattara.

**Writing – original draft:** Jean-Louis Kouakou, Sery Gonedelé Bi, Eloi Anderson Bitty, Célestin Kouakou, Alphonse Kouassi Yao, Kouadio Bénoîtype Kassé, Soulemane Ouattara.

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
