## [Decision Letter · Decision Letter 0]

15 Jun 2020

PONE-D-20-09118

Ivory Coast without ivory: massive extinction of African forest elephants in Côte d’Ivoire

PLOS ONE

Dear Dr. Gonedelé BI,

Thank you for submitting your manuscript to PLOS ONE. After careful consideration, we feel that it has merit but does not fully meet PLOS ONE’s publication criteria as it currently stands. Therefore, we invite you to submit a revised version of the manuscript that addresses the points raised during the review process.

We look forward to receiving your revised manuscript.

Kind regards,

Bi-Song Yue, Ph.D

Academic Editor

PLOS ONE

3. In your Methods section, please state where the people interviewed were recruited for your study.

4. Thank you for stating the following in your Ethics Statement that "Prior to the surveys, we received research permits from SODEFOR (Society of Forest Development) and OIPR (Ivorian Office of Parks and Reserves), respectively in charge of the management of Côte d’Ivoire’s protected areas. The study was reviewed and approved by SODEFOR and OIPR."

We noted that in your methodology you report that "Data on elephant distribution were also collected through structured interviews. Interviews with key informants from the surrounding local communities of PAs as well as field staff of the Forest and Wildlife Department" (lines 107-110).

For research involving human participants, such as conducting face-to-face interviews, we would expect approval from your institutional review board (IRB) or equivalent ethics committee(s) and reporting of participant consent. We would like to know whether all the research reported in your manuscript, in particular the work with human participants, has been performed at all times with ethical oversight by an ethics committee. Could you please clarify if your research permits included ethics oversight for conducting the face-to-face interviews?

If ethical approval was not required, please provide a clear statement of this and the reason why, and any relevant regulations under which the work is exempt from the requirement for approval. If the ethics approval was waived by your ethics committee, please provide a copy of this documentation formally confirming that ethical approval was not needed in this case, in the original language and in English translation as supporting information files. Please note that an email from your ethics committee will suffice. Please note that this is for internal use only and will not be published.

In addition, please provide additional details regarding participant consent. In the ethics statement in the Methods and online submission information, please ensure that you have specified (1) whether consent was informed and (2) what type you obtained (for instance, written or verbal). If the need for consent was waived by your ethics committee, please include this information.

Thanks for your attention to our requests.

5. Please upload a new copy of Figure 2 as the detail is not clear. Please follow the link for more information: https://blogs.plos.org/plos/2019/06/looking-good-tips-for-creating-your-plos-figures-graphics/"" https://blogs.plos.org/plos/2019/06/looking-good-tips-for-creating-your-plos-figures-graphics/" https://blogs.plos.org/plos/2019/06/looking-good-tips-for-creating-your-plos-figures-graphics/

Reviewers' comments:

Reviewer's Responses to Questions

**Comments to the Author**

1. Is the manuscript technically sound, and do the data support the conclusions?

Reviewer #1: Yes

Reviewer #2: Partly

2. Has the statistical analysis been performed appropriately and rigorously? 

Reviewer #1: Yes

Reviewer #2: No

3. Have the authors made all data underlying the findings in their manuscript fully available?

Reviewer #1: Yes

Reviewer #2: Yes

4. Is the manuscript presented in an intelligible fashion and written in standard English?

Reviewer #1: No

Reviewer #2: Yes

5. Review Comments to the Author

Reviewer #1: See attached

I've attached my review as indicated.

Reviewer #2: Questions/Issues

1) How did you differentiate forest from savanna elephant dung or are you just assuming that there are no savanna elephants in the study area?

2) Did you correct for the number of correlations run and the likely lack of independence between some of your variables? I also question some of your signs and p-values for your correlations; it would be helpful to see the sample sizes.

3) I’m confused by the locations where elephants were found. The abstract indicates in 5 PAs (34). The results state 5 PAs and 3 unprotected areas (166). The conclusion states 4 PAs (384) and says nothing about the unprotected areas.

4) The Discussion is overly long. The paper becomes a history lesson and a review. There are 7 pages of discussion compared to two pages of introduction. While it is preferred to have a longer discussion than introduction, much of the discussion goes well beyond the data of the paper. It could easily be trimmed by several pages by reducing repetition and tightening the message. The 96 citations is perhaps a bit excessive for the depth of the article.

Minor Edits

22 delete “further” – since no numbers are given in 19-20 it is impossible for the reader to know if the value provided in 21 are “one of the largest” or a decline.

24 The authors should say something about savanna elephants given the previous lines

31 You do not investigate with a statistical test – reword

36 Reword – improper grammatically – the Pas are not inhabiting elephants

37 Reword “largely reduced”

39 What are antipoaching actions?

46 They are two species (https://www.nature.com/articles/news.2010.691;
https://news.mongabay.com/2018/08/forest-elephant-dna-diverse-consistent-and-distinct-study-says/;
https://www.pnas.org/content/115/11/E2566;
https://will.illinois.edu/longerlisten/story/the-african-elephant-is-actually-two-separate-species-and-in-danger;
https://www.nature.com/articles/ng1485) any debate is political / legislative and not based on scientific evidence.

48 either “the elephant is” or “elephants are”

55 missing a period

58 last three

60 delete “poses a further threat” – unnecessary

67 in decline

79 delete “has”

83 change “of” to “for”

84 populations

86 change “paper” to “study”

87 change “get” to “obtain”

95 place a colon after “items” and delete “firstly”

96 place comma after “Parks” and after “restriction”

106 unclear if you mean “sign,” such as indicators of elephants, or “sign and interview surveys” in which case I am not sure of the meaning of “sign surveys” – use the Oxford comma if the former and explain if the latter

109 Interviews were conducted with …

112 “in the study” and “elephants in” (delete “from”)

114 “close to”

115 what’s the relevance of talking to researchers who do work at “other” places – clarify (e.g., bordering areas, etc.)

You may wish to look at this reference too: Youldon et al. 2017. Patch - occupancy survey of elephant (Loxodonta africana) surrounding Livingstone, Zambia. Koedoe, 59(1), a1372. https://doi.org/10.4102/koedoe.v59i1.1372.

136 You’ve already presented the (PA) so just use the PA

145 Again, you do not investigate with a statistical test

168 change “are” to “were”

186-8 Were both correlations positive? It would seem that elephant presence and poaching index would be opposite in sign to presence of elephants and level of protection of the PA. In 132 you define the poaching index such that the higher the sign of poaching the higher the index value.

191 So as the presence of elephants went up, the density went down (yet there was no correlation with size of the PA) …seems odd.

193-5 The lack of a significant correlation for these two measures is interesting and perhaps unexpected.

217 Reword – see previous comment about this structure

234 insert “and” before “increasing”

239 I’m not sure what “conducting to the illegal plantation” means

242 Do you mean “risen”?

299 change “does” to “do”

330 Use Oxford comma if journal permits

Continue to check grammar / sentence structure / word use through the end of the manuscript.

6. PLOS authors have the option to publish the peer review history of their article (what does this mean?). If published, this will include your full peer review and any attached files.

Reviewer #1: No

Reviewer #2: No

---

## [Author Response · Author response to Decision Letter 0]

31 Aug 2020

Adressed

Added as supporting information

3. In your Methods section, please state where the people interviewed were recruited for your study.

The people interviewed were recruited among local communities in the localities surrounding the surveyed PAs.

4. Thank you for stating the following in your Ethics Statement that "Prior to the surveys, we received research permits from SODEFOR (Society of Forest Development) and OIPR (Ivorian Office of Parks and Reserves), respectively in charge of the management of Côte d’Ivoire’s protected areas. The study was reviewed and approved by SODEFOR and OIPR."

We noted that in your methodology you report that "Data on elephant distribution were also collected through structured interviews. Interviews with key informants from the surrounding local communities of PAs as well as field staff of the Forest and Wildlife Department" (lines 107-110).

For research involving human participants, such as conducting face-to-face interviews, we would expect approval from your institutional review board (IRB) or equivalent ethics committee(s) and reporting of participant consent. We would like to know whether all the research reported in your manuscript, in particular the work with human participants, has been performed at all times with ethical oversight by an ethics committee. Could you please clarify if your research permits included ethics oversight for conducting the face-to-face interviews?

If ethical approval was not required, please provide a clear statement of this and the reason why, and any relevant regulations under which the work is exempt from the requirement for approval. If the ethics approval was waived by your ethics committee, please provide a copy of this documentation formally confirming that ethical approval was not needed in this case, in the original language and in English translation as supporting information files. Please note that an email from your ethics committee will suffice. Please note that this is for internal use only and will not be published.

Ethical approval in our country is not necessarily required for study dealing with animals. Indeed in appendix I of the law n ° 94-442 of August 16, 1994 amending the law n ° 65-255 of August 4, 1965 relating to the protection of fauna and hunting practices, fully protected wild animals including the elephant are prohibited from hunting and capture. Our study fully respected Ivorian regulations in force because it did not in any way affect the integrity of elephants nor the disturbance of that integrity. The authorizations issued by the authorities in charge of the protection of protected areas serve as their agreement regarding the regulations in force in our country.

In addition, please provide additional details regarding participant consent. In the ethics statement in the Methods and online submission information, please ensure that you have specified (1) whether consent was informed and (2) what type you obtained (for instance, written or verbal). If the need for consent was waived by your ethics committee, please include this information.

Before the interview, we explained the research purpose to the informants. We received a verbal consent of their acceptance to participate in the study.

Thanks for your attention to our requests.

5. Please upload a new copy of Figure 2 as the detail is not clear. Please follow the link for more information: https://blogs.plos.org/plos/2019/06/looking-good-tips-for-creating-your-plos-figures-graphics/" https://blogs.plos.org/plos/2019/06/looking-good-tips-for-creating-your-plos-figures-graphics/

Done

Reviewers' comments:

Reviewer's Responses to Questions

Comments to the Author

1. Is the manuscript technically sound, and do the data support the conclusions?

Reviewer #1: Yes

Reviewer #2: Partly

2. Has the statistical analysis been performed appropriately and rigorously?

Reviewer #1: Yes

Reviewer #2: No

3. Have the authors made all data underlying the findings in their manuscript fully available?

Reviewer #1: Yes

Reviewer #2: Yes

4. Is the manuscript presented in an intelligible fashion and written in standard English?

Reviewer #1: No

Reviewer #2: Yes

 5. Review Comments to the Author

Reviewer #1: See attached

I've attached my review as indicated.

We made corrections as indicated by the reviewer.

Reviewer #2: Questions/Issues

1) How did you differentiate forest from savanna elephant dung or are you just assuming that there are no savanna elephants in the study area?

The PAs surveyed extend in the southern zone of Côte d’Ivoire, the historical range of forest elephants, thus we assume that the population surveyed are forest elephants.

2) Did you correct for the number of correlations run and the likely lack of independence between some of your variables? I also question some of your signs and p-values for your correlations; it would be helpful to see the sample sizes.

The correlation between the presence of elephant and site variables (size of the forest, percentage of area converted into plantation, size of the forest left, size of human population inside the PA, poaching index, distance to the nearest road, population density in the Department, level of protection of the PA) was evaluated with Pearson correlation. These variables were log transformed as needed to improve normality. In total 25 PAs were surveyed.

3) I’m confused by the locations where elephants were found. The abstract indicates in 5 PAs (34). The results state 5 PAs and 3 unprotected areas (166). The conclusion states 4 PAs (384) and says nothing about the unprotected areas.

Elephants were found in 4 PAs and 3 unprotected areas. This has been corrected in the document.

4) The Discussion is overly long. The paper becomes a history lesson and a review. There are 7 pages of discussion compared to two pages of introduction. While it is preferred to have a longer discussion than introduction, much of the discussion goes well beyond the data of the paper. It could easily be trimmed by several pages by reducing repetition and tightening the message. The 96 citations is perhaps a bit excessive for the depth of the article.

The discussion section has been reduced. Efforts were made to reduce repetition. Citations have been reduced to 57.

Minor Edits

22 delete “further” – since no numbers are given in 19-20 it is impossible for the reader to know if the value provided in 21 are “one of the largest” or a decline.

Done

24 The authors should say something about savanna elephants given the previous lines

As the study focus on forest elephant we think that there is no need to say more on savannah elephant in the abstract.

31 You do not investigate with a statistical test – reword

We used Pearson correlation to determine the correlation between the presence of forest elephant and site variables

36 Reword – improper grammatically – the Pas are not inhabiting elephants

PAs with higher level of protection have higher probability to be home of elephant population.

37 Reword “largely reduced”

The viability of these populations is uncertain, since they have a small size and are isolated.

39 What are antipoaching actions?

Antipoaching actions refer here to ranger patrolling

46 They are two species (https://www.nature.com/articles/news.2010.691;
https://news.mongabay.com/2018/08/forest-elephant-dna-diverse-consistent-and-distinct-study-says/;
https://www.pnas.org/content/115/11/E2566;
https://will.illinois.edu/longerlisten/story/the-african-elephant-is-actually-two-separate-species-and-in-danger;
https://www.nature.com/articles/ng1485) any debate is political / legislative and not based on scientific evidence.

Genetic evidence suggests two distinct species among African elephants: the forest elephant (Loxodonta cyclotis) and the savanna elephant (L. africana).

48 either “the elephant is” or “elephants are”

Corrected

55 missing a period

In pre-colonial times (before 1893)

58 last three

Corrected

60 delete “poses a further threat” – unnecessary

Deleted

67 in decline

These populations are confined to protected areas and continue to decline

79 delete “has”

Deleted

83 change “of” to “for”

Done

84 populations

Corrected

86 change “paper” to “study”

Done

87 change “get” to “obtain”

Done

95 place a colon after “items” and delete “firstly”

Protected areas are categorized into National Parks which normally are under strong protection restriction, and Forest Reserves that are partially protected.

96 place comma after “Parks” and after “restriction”

Done

106 unclear if you mean “sign,” such as indicators of elephants, or “sign and interview surveys” in which case I am not sure of the meaning of “sign surveys” – use the Oxford comma if the former and explain if the latter

sign, and interview surveys… Sign as indicators of elephants

109 Interviews were conducted with …

Corrected

112 “in the study” and “elephants in” (delete “from”)

Done

114 “close to”

Corrected

115 what’s the relevance of talking to researchers who do work at “other” places – clarify (e.g., bordering areas, etc.)

You may wish to look at this reference too: Youldon et al. 2017. Patch - occupancy survey of elephant (Loxodonta africana) surrounding Livingstone, Zambia. Koedoe, 59(1), a1372. https://doi.org/10.4102/koedoe.v59i1.1372.

We also talked to researchers who conducted scientific or conservation work in these areas and the bordering areas

136 You’ve already presented the (PA) so just use the PA

Done

145 Again, you do not investigate with a statistical test

………..was evaluated with Pearson correlation

168 change “are” to “were”

Done

186-8 Were both correlations positive? It would seem that elephant presence and poaching index would be opposite in sign to presence of elephants and level of protection of the PA. In 132 you define the poaching index such that the higher the sign of poaching the higher the index value.

Our data indicated no significant correlations between the presence of elephants in the PAs and poaching index (r = 0.223, p = 0.283). This could be explained by the fact that all the poaching signs are not necessarily targeting elephants. 

191 So as the presence of elephants went up, the density went down (yet there was no correlation with size of the PA) …seems odd.

There is no correlation between the presence of elephant and the size of the PA. As well as the PA has a large size, once there is no protection and that the habitat of that PA is degraded, there is less chance to find elephants.

193-5 The lack of a significant correlation for these two measures is interesting and perhaps unexpected.

The lack of significant correlation between the presence of elephants and the distance to the nearest road (r = - 0.176, p = 0.399) or the size of the PA (r = 0.217, p = 0.298) is effectively unexpected and indicates the level of disturbance that are facing the surveyed PAs. 

217 Reword – see previous comment about this structure

Done

234 insert “and” before “increasing”

Done

239 I’m not sure what “conducting to the illegal plantation” means

Conversion of protected areas into plantations

242 Do you mean “risen”?

Yes

299 change “does” to “do”

Done

330 Use Oxford comma if journal permits

Not changed, to lengthening the sentence

Continue to check grammar / sentence structure / word use through the end of the manuscript.

We make effort to check grammar / sentence structure / word use through the end of the manuscript.

6. PLOS authors have the option to publish the peer review history of their article (what does this mean?). If published, this will include your full peer review and any attached files.

Do you want your identity to be public for this peer review? For information about this choice, including consent withdrawal, please see our Privacy Policy.

Reviewer #1: No

Reviewer #2: No

Done

---

## [Editor Report · Decision Letter 1]

10 Sep 2020

Ivory Coast without ivory: massive extinction of African forest elephants in Côte d’Ivoire

PONE-D-20-09118R1

Dear Dr. Gonedelé BI,

We’re pleased to inform you that your manuscript has been judged scientifically suitable for publication and will be formally accepted for publication once it meets all outstanding technical requirements.

Kind regards,

Bi-Song Yue, Ph.D

Academic Editor

PLOS ONE

---

## [Editor Report · Acceptance letter]

17 Sep 2020

PONE-D-20-09118R1 

Ivory Coast without ivory: massive extinction of African forest elephants in Côte d’Ivoire 

Dear Dr. Gonedelé Bi:

I'm pleased to inform you that your manuscript has been deemed suitable for publication in PLOS ONE. Congratulations! Your manuscript is now with our production department. 

Kind regards, 

on behalf of

Dr. Bi-Song Yue 

Academic Editor

PLOS ONE